# Constructing a Comprehensive National Wildfire Database from Incomplete Sources: Israel as a Case Study

Edna Guk [1,2,*], Avi Bar-Massada [3] and Noam Levin [1,4]

1 Department of Geography, The Hebrew University of Jerusalem, Jerusalem 9190501, Israel
2 Israel Nature and Parks Authority, 3 Am Ve Olamo Street, Jerusalem 9546303, Israel
3 Department of Biology and Environment, University of Haifa at Oranim, Kiryat Tivon 36006, Israel
4 Remote Sensing Research Centre, School of Earth and Environmental Sciences, University of Queensland, Brisbane, QLD 4072, Australia
* Correspondence: edna.guk@mail.huji.ac.il

**Abstract:** In many regions, the frequency and extent of wildfires has increased in recent years, a trend which is expected to continue. Hence, there is a need for effective fire management strategies. Such strategies need to be based on accurate and complete data on vegetation condition and post-fire effects, collected in the field as well as by remote sensing approaches. Unfortunately, wildfire databases are often incomplete in terms of their spatial and temporal coverage, as well as the documentation of fire outcomes. The aim of this study was to devise a methodology to construct a comprehensive national wildfire database. We developed and tested this methodology for Israel, as a case study. The methodology combines data on wildfires in Israel from two sources: remote sensing and field data collected by governmental agencies, representing the period between 2015 and 2022. The resulting database contains 2276 wildfires larger than 10 ha with information (where and when available) on their occurrence date, fire duration, ignition cause, and more. The methodology presented here provides a blueprint for developing large-scale and comprehensive fire databases elsewhere, and facilitates future studies on wildfire risk by providing a robust, unified database of the fire history of Israel from 2015 onwards.

**Keywords:** wildfires; VIIRS; MODIS; Sentinel-2; Landsat-8; database; RDNBR

## 1. Introduction

### 1.1. Remote Sensing of Wildfires

Wildfires are a widespread global phenomenon that is prevalent in many ecosystems. Land use change processes such as afforestation, settlement development in or near vegetated ecosystems, and changes in agricultural practices may have a considerable influence on the amount, composition, and configuration of vegetative fuels, and alter the spatial and temporal patterns of fire ignitions and spread. In tandem with land use and land cover changes in many regions in recent decades, there has been an increase in fire frequency worldwide [1–4]. However, Andela et al. [5] found that across ecosystems and human land management types, fire characteristics such as fire size, ignition type, duration, and rate of spread varied widely. In general, the trend of an increase in fire frequency is expected to continue with global warming and additional land use/land cover changes [4,6–10].

To better understand changes in fire regimes and the threat of wildfire to both humans and natural ecosystems, it is necessary to have long-term data on the occurrence, size, and characteristics of fires, i.e., detailed fire atlases. These atlases typically combine information from two types of data sources: field measurements and remote sensing. The former often consists of in situ identification of the ignition point and its cause, and the latter allows the delineation of the burned area and the estimation of fire severity. As these measurements are difficult to obtain in situ (especially for large fires) and are limited in their ability to

reconstruct past fire events, remote sensing products have become a major tool for creating fire atlases, especially where field data are lacking [11–15].

Satellites with sufficient spatial, temporal, and radiometric resolution along with or without field sampling are common tools for mapping active fires, fire scars, and burn severity, especially in inaccessible areas [16–19]. Furthermore, satellites are useful for studying fire impacts on ecosystems and for mapping environmental characteristics such as fuel moisture content, vegetation type, slope, aspect, and anthropogenic variables (e.g., proximity to roads and settlements) that facilitate the mapping of fire risk [20–23]. The common satellites for mapping fire occurrence, extent, and severity as well as variables needed for fire risk assessment are MODIS, VIIRS Suomi NPP, and NOAA-20 (formerly JPSS-1), Landsat, and Sentinel-2. Their spatial resolution is 250–1000, 375–750, 30, and 10 m/pixel, and their revisit times are 1–2 days, daily, 16 days (till 2021), and 5 days, respectively [24–26]. Their spectral bands range between the visible, Near-Infrared (NIR), Short-Wave Infrared (SWIR), and Thermal Infrared (TIR) wavelengths. The varying characteristics of different satellite sensors highlight their potential to act as complementary data sources for fire atlases.

In the context of fire atlases, two remote sensing data products are specifically important: active fires (which identify fires as they occur) and burned areas (which retrospectively delineate the outcomes of fires in terms of burn scars). The daily live hotspot fire products from MODIS and VIIRS, based on thermal anomalies [27,28] (https://firms.modaps.eosdis.nasa.gov/map/, accessed on 5 July 2021), are the predominant active fire products. Numerous studies showed that after masking urban areas, active fires (hot spot) products from MODIS-1 km had low commission errors, but high omission errors in detecting wildfires. The omission error percent varied widely, depending mainly on fire size (it improved when the fire size threshold was higher), but was also influenced by the land cover type, cloudiness, canopy density, and seasonality [29,30]. VIIRS 375 m S-NPP significantly reduces omission errors compared to the MODIS 1 km data product and exhibits a high correspondence with Landsat-8 reference data. As to commission errors, both sensors were influenced by the derived fire radiative power values and the land cover types [28,31,32]. For instance, Giglio et al. [33] showed that MODIS had the lowest probability of wildfire detection in the Middle East, 7% (for a median size of ~3600 m$^2$), compared to a maximum of 26% of the fires (for a median size of ~24,300 m$^2$) in Australia and New Zealand.

For retrospective mapping of burned areas, various indices were developed such as the Normalized Burn Index (NBR) [34], the Burned Area Index (BAI) [35], and the Relative Difference Aerosol-Free Vegetation Index (RdAFRI) [36]. Moreover, for improved separation between burned and unburned pixels and for assessing fire severity, various indices were developed based on change detection. DNBR, along with Relative DNBR (RDNBR), are among the most widely used spectral indices for mapping burn severity from remote sensing [37]. Burn severity refers to the effect of fire on the ecosystem (soil and vegetation) [34]. All remote-sensing based measures of burn severity are ultimately limited as they rely on detecting changes in the spectral properties of vegetation as an indirect proxy for their structure and function. In contrast, ground measurements can provide detailed information on burn severity, but they are restricted to small areas because they rely on complex and expensive sampling processes. Hence, to complement the different limitations of each approach, ground measurements and remote sensing must be combined for validation and calibration in order to correctly assess the severity of a wildfire [38]. Recently, research conducted on forest fires in Israel that compared burn severity assessment by visual and spectral indices showed that the spectral indices can effectively separate burned and unburned areas and distinguish between high and medium-high severity of the forest fire [39].

*1.2. Wildfire Database Creation from Remote Sensing and Field Data*

There are many wildfire datasets (or fire atlases) worldwide at different scales, regions, and formats [16,40–43]. For example, Andela et al. [14] developed an algorithm for creating

the Global Fire Atlas Database, freely available as a GIS shapefile layer through https://daac.ornl.gov/cgi-bin/dsviewer.pl?ds_id=1642 (accessed on 2 January 2022). This database includes daily information on individual fires (≥21 ha, from 2003 to 2016), on burned land use type, and on seven fire characteristics across all biomes: ignition timing and location, fire size, duration, daily expansion, daily fire line, speed, and direction of spread. The data are derived from the Global Fire Atlas Algorithm and MODIS Collection 6 MCD64A1 burned area product at 500 m resolution [14]. Additionally, Laurent et al. [42] created FRY, a global database of fire patches based on MCD64A1 Collection 6 and the MERIS fire_cci v4.1 datasets, for the years 2005–2011. The database is available in the OSU-OREME repository. In addition, Artés et al. [43] created the GlobFire Database of wildfires (also based on MODIS burnt area product Collection 6, MCD64A1) that focused on the characterization of fire types and fire regimes. The database is available in the Global Wildfire Information System (GWIS) platform. Since the global burned area database from MODIS is nearing the end of its life cycle, new global burned area databases with lower omission error were developed from Sentinel-3 and VIIRS active fire at 300 m resolution [44]. The European Forest Fire Information System (EFFIS) is another example of an international forest and uncontrolled vegetation fire database, covering over 30 European and neighbouring countries. EFFIS supplies a complete cycle of forest fire management, from before to post-fire damage analysis [40]. EFFIS includes the European Fire Database, yearly reports, web applications, and post-fire assessment [40].

Unfortunately, global fire datasets are not relevant for regions where most fires are small and have a short duration, as they go undetected by NASA's active fire products [33,45]. Taking Israel as an example, the global fire atlas [14] for the years 2015 and 2016 had only 37 and 38 wildfires, with a median area of 21 and 42 ha and a total burned area of 1759 and 2981 ha, respectively. The fire regime in Israel, where vegetative fuels are highly fragmented, is characterized by a mean burned area per month of about 11 to 33 ha (depending on month, based on the INPA wildfire database), and an average duration of less than one day [46,47]. Still, these fires cause significant damages to property and ecosystems, and pose risk to human lives. Therefore, there is urgent need to obtain consistent and comprehensive information about their causes, their ignition locations, and their consequences in terms of burned areas and fire severity.

In many countries, there are different agencies in charge of data collection on wildfire occurrence, spread, and outcomes, often due to fragmented land ownership patterns [48]. In Israel, specifically, there are profound differences in fire data documentation among the three authorities involved in managing open spaces: The Jewish National Fund (JNF, Israel's Forestry Service), The Israel Nature and Parks Authority (INPA), and Israel's National Fire and Rescue Authority (IFRS) [49]. Consequently, the documentation of wildfires is not systematic and uniform [49]. In addition, Levin et al. [47] showed differences in the spatial distribution of wildfires from the JNF point database (that were biased towards small fires) and wildfire scars derived from MODIS (that were biased towards large fires). Consequently, there are many knowledge gaps regarding the spatial and temporal distribution of fires and fire risk in general. For effective fire prevention and management at the local scale, there is a need for a unified, accurate, and validated database [48]. Our study was motivated by this need. Given the limited and disorganized state of much of the current fire data generated by different governmental agencies in Israel, we suggest that remote sensing approaches may be able to supplement existing field data from these different sources, and facilitate the development of a consistent national fire database. While our case study focuses on Israel and depends on the particularities of its existing fire databases, our study can provide useful insights into the development of fire atlases in other countries as well.

### 1.3. Research Aim

Our aim was to develop a methodology for creating a unified database of wildfires by combining remote sensing algorithms with historical fire data from governmental agencies.

We exemplify our approach in Israel, using four remote sensing datasets: two active fires datasets (MODIS and VIIRS) and two time series of landsat-8 and sentinel-2. We also use three independent institutional data sources. Creating a unified database is a crucial step for achieving a better understanding of Israel's fire regime.

Hypotheses:

- VIIRS and MODIS active fires products have high omission error percentages in detecting wildfires in Israel due to the prevalence of fires whose area is small and temporal duration is short.
- Wildfires are a multi-dimensional and complex phenomenon, and therefore several complementary variables are needed to effectively describe a given wildfire event.
- There would be a positive correlation between a wildfire's burned area and its likelihood of inclusion in multiple independent fire databases.

## 2. Materials and Methods

### 2.1. Study Area

In Israel as in many other regions, the frequency and extent of wildfires has increased in recent years [47]. The wildfire season in Israel coincides with the dry season, which usually takes place between April and November with maximum average temperatures of ~33 degrees Celsius in July and August [50]. The precipitation gradient in our study area, the Mediterranean region of Israel (which does not include the desert region south of Beer Sheva as there is almost no vegetation there) ranges from ~300 to ~1300 mm/y. The size of wildfires in Israel is typically small (the majority are smaller than 10 ha), and their duration is short, as forest, woodland, and shrubland patches in Israel are relatively small and highly fragmented [46]. The areal cover of natural areas and forests northern to Beer Sheva is 4300 km2, and 3710 km2 are covered by agriculture areas [51]. The main cause of wildfire ignitions in Israel is human activity, either unintentionally [47] or intentionally via arson [36,52]. There are virtually no natural fires in Israel as there are almost no lightning storms in the summer, and only six cases of lightning fire ignitions were ever documented in Israel [53]. Current drivers of fire initiation and spread include an increase in fuel amount and continuity due to a long-term decrease in grazing pressure and the subsequent regeneration of Mediterranean woodlands, the expansion of forest plantations, negligence by human actors such as visitors in forests, military units in training, arson; and agricultural fires that accidentally start by farmers who burn agricultural waste [47,54,55].

### 2.2. Data Sets

#### 2.2.1. Governmental Agencies

The wildfire databases of governmental agencies in Israel were independently generated by each agency using different methods, each with its advantages and limitations. The Jewish National Fund (JNF) database consists of a polygon layer of fire scars from 2017 created by field sampling, and includes the fire date, burned area, fire type, and additional comments. The data are mostly biased toward JNF's management zones (mostly planted forests), but also include small wildfires in other areas due to the collection method. Before 2017, the wildfire database of JNF consisted of centroids of wildfire events [47]. Israel's Nature and Parks Authority (INPA) has two separate databases. The first consists of polygons of fire scars created by manual digitization from Sentinel-2 and Landsat satellite images at bi-weekly intervals since 2015. The second INPA database is collected in situ by rangers participating in the fire suppression efforts, using a designated smartphone app. The app allows INPA rangers to report directly from the field on various types of information, including animal observations, habitat surveys, the presence of hazards, wildfires and more, using GPS and mapping tools. This INPA wildfire database contains a polygon layer of fire scars with additional complementary information as shown in Figure 1. However, the number of wildfires in INPA's database which is derived from rangers' reports is limited and includes only a subset of Israel's wildfire events each year (only when an INPA ranger participated in wildfire suppression efforts, and mainly in nature reserves and national

parks). Finally, the Israel Fire and Rescue Services (IFRS) database consists of a point layer of all fire events to which at least one firetruck was dispatched. The IFRS was only established in 2013, and this database is only available since 2013, with data from earlier years being much less accurate compared to recent years. The IFRS database also contains complementary information about the fire, as shown in Figure 1. The significant limitation of the IFRS database is that the location of the ignition point is not always accurate, and the database includes not only wildfires but also house fires that were erroneously classified as wildfires.

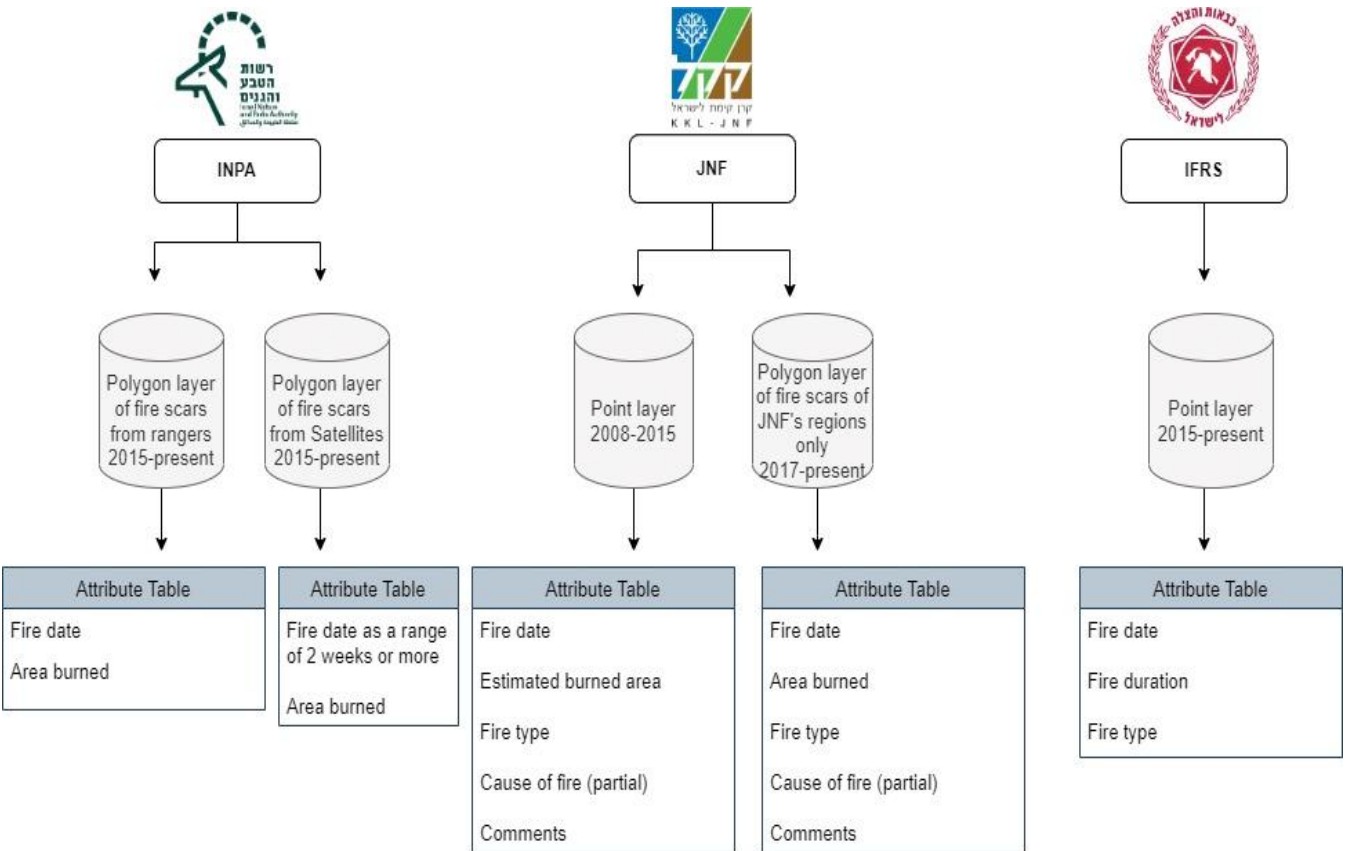

**Figure 1.** The datasets used to create Israel's national wildfire database from three main organizations that collect wildfire data in Israel. INPA-Israel Nature and Parks Authority; JNF-Jewish National Fund; IFRS-Israel Fire and Rescue Services.

### 2.2.2. Remote Sensing Data

The INPA database was verified and combined with available remote sensing products. The remote sensing datasets we used included the following: (1) active fire points from MODIS and VIIRS with information on accurate wildfire occurrence date and the fire radiative power, and (2) various spectral indices which we calculated from time series of Landsat-8 (2015–2022) and Sentinel-2 (2018–2022) images: RDNBR, NDVI, and DNDVI (as detailed below). As the differences in pre-fire chlorophyll content among different vegetation types and densities cause a biasing effect of the pre-fire condition, we used the RDNBR to enable a more direct comparison across landscapes and time [34,56,57]. DNBR/RDNBR has negative values when vegetation re-establishes in a site, such as herbaceous communities that respond quickly, while high positive values are assumed to reflect post-fire sites with decreased productivity such as forested and shrub-dominated areas [34,56]. In addition, we calculated NDVI [58,59] for assessing the vegetation condition before the wildfire and DNDVI for assessing the change in vegetation before and after a wildfire event (Figure 2). Figure 3 presents an example of available governmental and

remote sensing databases for northern Israel, 2020. The spectral indices we used in the analysis are the following:

1. NBR = (NIR − SWIR)/(NIR + SWIR);
2. DNBR = NBR (pre fire) − NBR (after fire);
3. RDNBR = (pre fire NBR − after fire NBR)/$\sqrt{}$(ABS(pre fire NBR/1000));
4. NDVI = (NIR − RED)/(NIR + RED);
5. DNDVI = NDVI (pre fire) − NDVI (after fire).

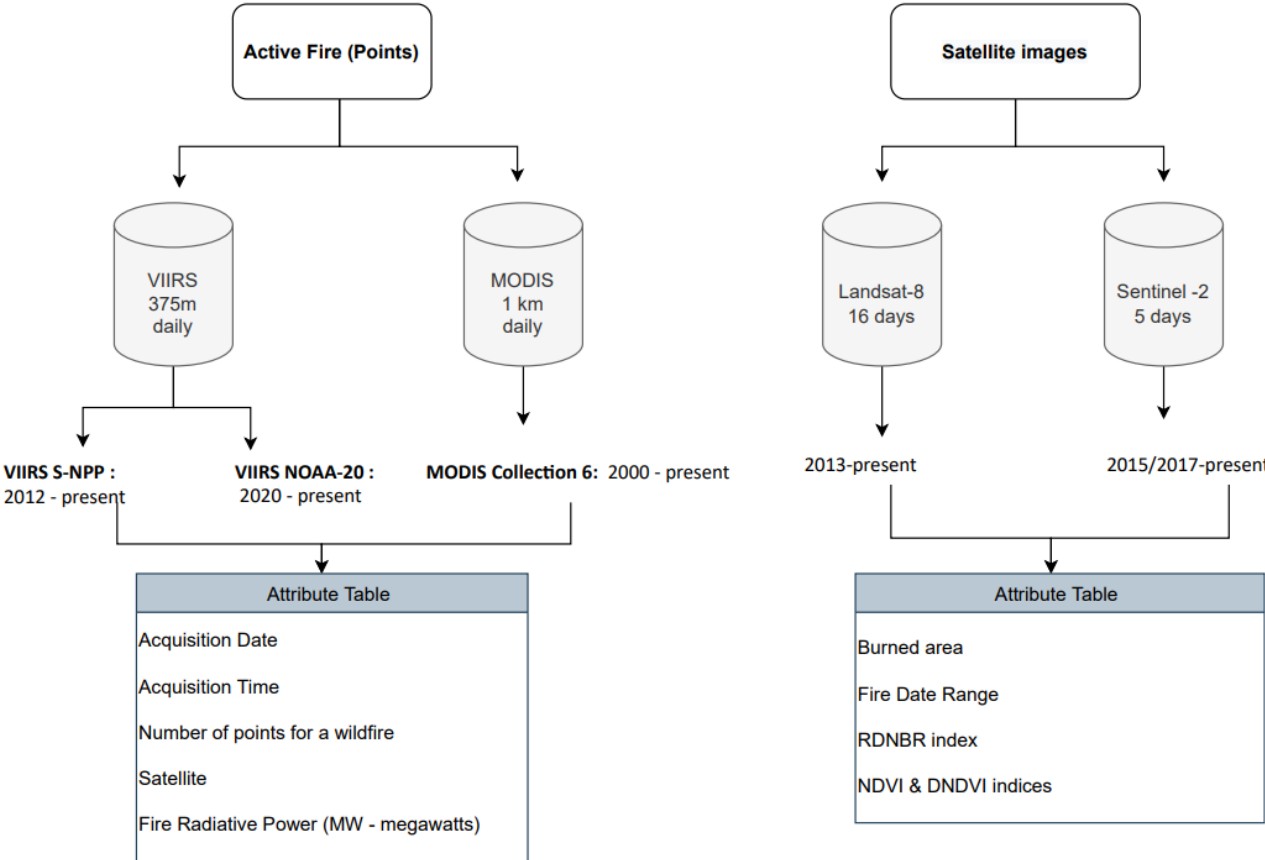

**Figure 2.** The remote sensing sources used for creating Israel's national wildfire database (https://earthdata.nasa.gov/earth-observation-data/near-real-time/firms, accessed on 5 July 2021); NASA Near Real-Time and MCD14DL MODIS Active Fire Detections (SHP format). Dataset available online (https://earthdata.nasa.gov/active-fire-data, accessed on 5 July 2021). The MOD14 Collection 6 daytime global commission error is 1.2% [33]; NASA Near Real-Time VNP14IMGTDL_NRT VIIRS 375 m Active Fire Detections (SHP format). Datasets available online (https://earthdata.nasa.gov/active-fire-data, accessed on 5 July 2021).

*2.3. The Unified Database Creation Process*

Generally, there are commonalities among different databases of wildfires worldwide in terms of accuracy and validity. The fundamental challenge of merging different wildfire databases is to ensure that they refer to the same wildfire event despite potential differences in ignition date, ignition location, and burned area delineation. The correct identification of a given fire event in different data sources enables the extraction of complementary information and informs on the confidence that can be assigned to various fire characteristics in the resulting unified database. The second challenge in creating a unified database is ensuring record completion and correction, which reflects the following possible issues:

• Incompleteness in recorded wildfires events (omission errors);

- Spatial accuracy problems (mistakes in wildfire scar size, its boundary, and/or ignition location);
- Temporal accuracy problems (mistakes in a wildfire occurrence date and/or duration).

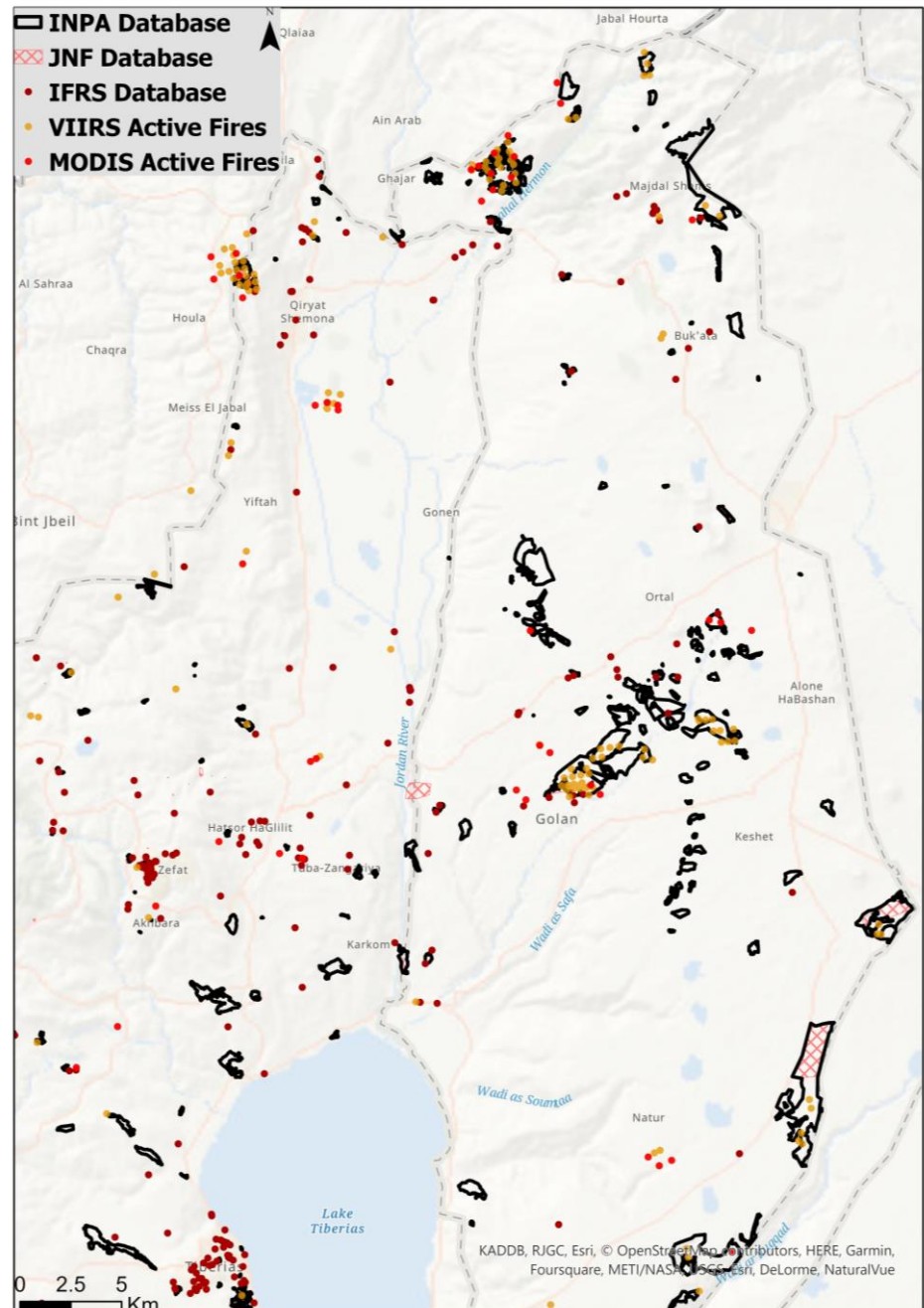

**Figure 3.** An example of records from the different wildfire databases used in this study (INPA, JNF, IFRS, and VIIRS and MODIS active fires). Data shown are for wildfires in northern Israel in 2020.

Our subsequent methodology, therefore, attempts to mitigate these issues by first identifying omissions in different databases, and then by merging the same wildfire records from multiple databases while extracting complementary information from them.

### 2.3.1. Identifying Missing Wildfires

First, we needed to decide which database would serve as our primary database and what would be the minimum burned area threshold that we will aim to cover in the national database. We chose the INPA wildfire database (polygon layer of fire scars

manually mapped from Landsat-8 (for the years 2015–2019) and Sentinel-2 (2020–2022) as it was the most complete among the three available options. Given that wildfires were identified based on visual inspection of pre-fire and post-fire images that are not separated by a single day, the date of a wildfire was not accurately determined, and was estimated to be within a range of two weeks. We decided that our national database will include only wildfires which are larger than 10 ha. This threshold value was chosen based on a visual inspection of burned area histograms for each year derived from the original INPA database (Figure 4). According to Gilgio et al. [33] the median fire size in the Middle East is about 0.4 ha (4 ASTER pixels); however, their probability of detection with MODIS is very low (~7%). In a study of fires in two regions of Spain and Portugal [60], the mean fire sizes reported were 59 and 101 ha (respectively), with more than 70% of the fires between 1 and 10 ha. However, more than 96% of the burnt area resulted from fires larger than 10 ha, highlighting the disproportional effect of fires larger than 10 ha on fire outcomes. In a recent study of wildfires in Lebanon, the minimum fire size properly captured by remote sensing approaches was 6.3 ha [45]. Following a similar approach, we identified for wildfires in Israel a threshold of 10 ha, which created a right-skewed distribution for all years that corresponds with the actual size distribution of wildfires, i.e., most wildfires are small and few are large, with a power law probability distribution [61,62]. If we chose a smaller threshold, the size distribution would be erroneously defined as right-skewed normal, as an artifact of missing a large number of small fires that go undetected even by visual image interpretation. Hence, our choice of a 10 ha threshold ensures that the atlas comprises high-confidence fire counts across all fire sizes above the threshold. Moreover, from a practical standpoint, focusing on larger burned areas increases the likelihood of detection by multiple remote sensing sources (with lower spatial resolution as Landsat-8 (2015–2018) compared to Sentinel-2 (2019–2022) (Figure 4), and subsequently improves the quality of the resulting fire atlas. Given that the manual wildfire mapping process in INPA was incomplete until early 2019, we re-examined time series of Landsat 8 images to identify additional wildfires which were missing in the original INPA database. We divided the study area into grid cells of 5 on 5 km, masked the wildfires that were already mapped in the original INPA database, and manually mapped the extent of missing wildfires by visual identification of fire scars on a false color composite imagery (NIR, RED, GREEN). This method was similar to the mapping method of the original INPA database for the years 2015–2018. As ploughed field could mistakenly be classified as wildfire, we did not map burn scars in croplands, and focused on mapping of wildfires in areas dominated by natural vegetation.

2.3.2. Improving the Temporal and Spatial Accuracy of Wildfires in the INPA Database

To validate and improve our estimate of the time a wildfire took place, we calculated in Google Earth Engine a time series of the mean RDNBR index and its rank for each wildfire polygon (in the time series of a given year, from April to November). This time series contained ~46 values per polygon for Sentinel 2 and ~15 values in the case Landsat-8 (depending on the availability of cloud free images). When the rank of the RDNBR index was high (at least one out of three highest values in a time series for each polygon) and within the date range for wildfire ignition from the INPA database, the occurrence date of the wildfire was set to the range between the date of the last satellite image before the wildfire and the first satellite image date after the wildfire (corresponding with the RDNBR index dates) (Figure A1, stage 1). When the rank of RDNBR was low (>3), the wildfire polygon was visually compared to a corresponding satellite image (Landsat 8/Sentinel-2), and its date was changed accordingly in one of the following three ways: (1) assigning a correct range of the wildfire dates; (2) the wildfire polygon was divided into several wildfire events which took place on different dates if it was found that fire scars from separate wildfire events were mistakenly identified as belonging to a single wildfire event in the original INPA database; or (3) removed (Figure A1, stage 1, and Figure A2).

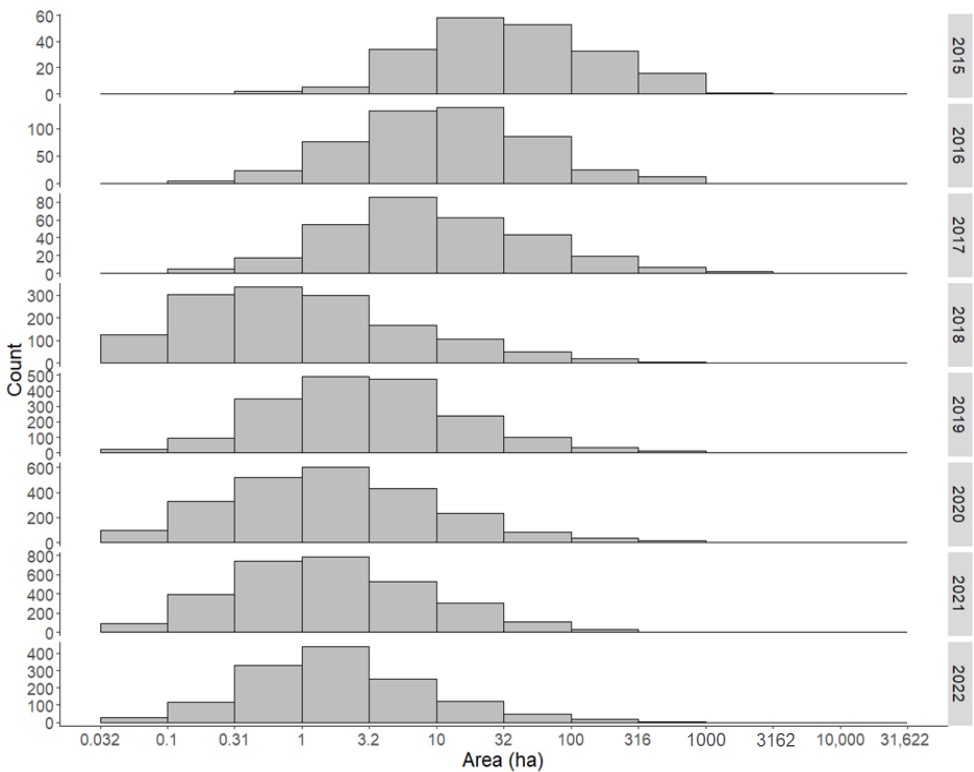

**Figure 4.** Distributions of burned areas between 2015 and 2022 (from INPA's original database). Fire size is shown on a logarithmic scale. The shift of the histograms of 2018–2022 to the left, in comparison to those of 2015–2017, is due to the improved spatial resolution of Sentinel-2 in comparison with Landsat.

2.3.3. Merging the Same Wildfire Event from Various Databases and Accuracy Assessment

Our next step was to merge information on wildfires from the different databases. Using R and ArcGIS Pro, the attribute data from different databases was joined to INPA polygons of wildfires by a certain order, and by different spatial and temporal conditions as described in Figure A1. The order of the different joined databases was chosen by their spatial and temporal accuracy and completeness. First, the active fires (from MODIS and VIIRS) with more reliable dates and locations were joined (Figure A1, stages 2, and 3). During this step we found seven missed wildfires between 2019 and 2020, and their extents were manually digitized from relevant sentinel-2 images. Only then did we join the attribute data from the JNF and IFRS datasets (Figure A1, stage 4 and 5). We estimated that the JNF had greater temporal uncertainty and IFRS datasets had both spatial and temporal uncertainty as they were collected and reported manually by people. Additionally, the spatial and temporal thresholds we used to merge the attributes of wildfire events between polygons and points of the wildfires from various databases were chosen based on the joined database's spatial resolution and accuracy. The temporal resolution threshold for identifying fire events from different datasets was set to ±1 day (given that only the largest fires in Israel last more than a single day, which happens only for very few fires every year), and the spatial threshold was set to 100 m between wildfire polygons of JNF and INPA (as their locations were pretty accurate because of the mapping method), 500 m for VIIRS active fires (as the spatial resolution is 375 m, though we slightly increased the distance due to possible spatial inaccuracies of smaller wildfires), and 1000 m for MODIS active fires (corresponding to pixel size) and the IFRS database (Figure A1). The numeric attributes of the final database are presented in Table 1.

**Table 1.** Acronym table of the numeric attributes of the final database.

| Acronym of the Numeric Attribute | Description |
| --- | --- |
| Avg FRP VIIRS, Max FRP VIIRS, Min FRP VIIRS | Average, maximum, and minimum value of Fire Radiative Power from VIIRS active points related to the same wildfire event (same date, and the distance between features <500 m). |
| Points num VIIRS | The number of active fire points from VIIRS that represent the same wildfire event. |
| Avg FRP MODIS, Max FRP MODIS, Min FRP MODIS | Average, maximum, and minimum value of Fire Radiative Power from MODIS active points related to the same wildfire event (same date, and the distance between features <1000 m). |
| Points num MODIS | The number of active fire points from MODIS that represent the same wildfire event. |
| Area (ha) | The burned area of a wildfire (based on INPA's database). |
| IFRS Duration | The duration of a wildfire in minutes, from the Israel Fire and Rescue Services (the time passed from the first firetruck was sent to the time the last firetruck left). |
| RDNBR L8/S2, Rank L8/S2 | Mean RDNBR index of each wildfire polygon from Landsat-8 (L8)/Sentinel-2 (S2), the rank of the index value in a time series. |
| Pre NDVI | The mean NDVI before the wildfire (from the last image without clouds). |
| DNDVI, Rank DNDVI | The difference between the NDVI index before and after the wildfire, the rank of the DNDVI in a time series. |

Independently validating our database is difficult because there is no available independent fire database in Israel with a finer spatial resolution for comparison. (In fact, the creation of such database was the main motivation for our study.) Consequently, we had to use an indirect approach to estimate the quality of the new dataset. To do so we rated the reliability of a given wildfire event by the number of independent databases in which it was included, as well as by its RDNBR rank. The reliability of a database record is considered high if it was found in multiple independent databases, and it is associated with a high RDNBR rank. Nevertheless, as an additional quality assessment step we also compared our national database with the global burned area product FireCCIS310 [44]. To facilitate direct comparison, we restricted it only for wildfires containing at least two pixels (as the threshold value in our database was 10 ha). The comparison was based on the number of wildfires (including missing wildfires) and total burned area by month in 2019 only (the relevant FireCCI year).

## 3. Results

### 3.1. Constructing the National Wildfire Database

3.1.1. Improving the Completeness of the INPA Database

The final database contained 2276 wildfires with a total burned area of 142,911 ha between 2015 and 2022. Of these wildfires, 167 wildfires larger than 10 ha with a total burned area of 5227 ha were missing from the original INPA wildfire database for 2015–2020. Additionally, we removed 75 polygons of wildfires larger than 10 ha (with a total burned area of 1409, mostly in agricultural areas) between 2016 and 2022, as they were mistakenly classified as wildfires. Figure 5 describes the number and area of wildfires that were removed and added compared to the original INPA database (from the refining process, Figure A1, stage 1) by year. In the earlier years (2015–2016), the number of wildfires that were missing was large compared to more recent years (2017–2018), and in the years 2019–2020 only a few wildfires were missing in the INPA database, and these were detected by the VIIRS. Overall, the average wildfire area was similar across all years. The smallest

mean wildfire area was found in 2019 (16 ha), and the highest mean wildfire area was found in 2020 (43 ha).

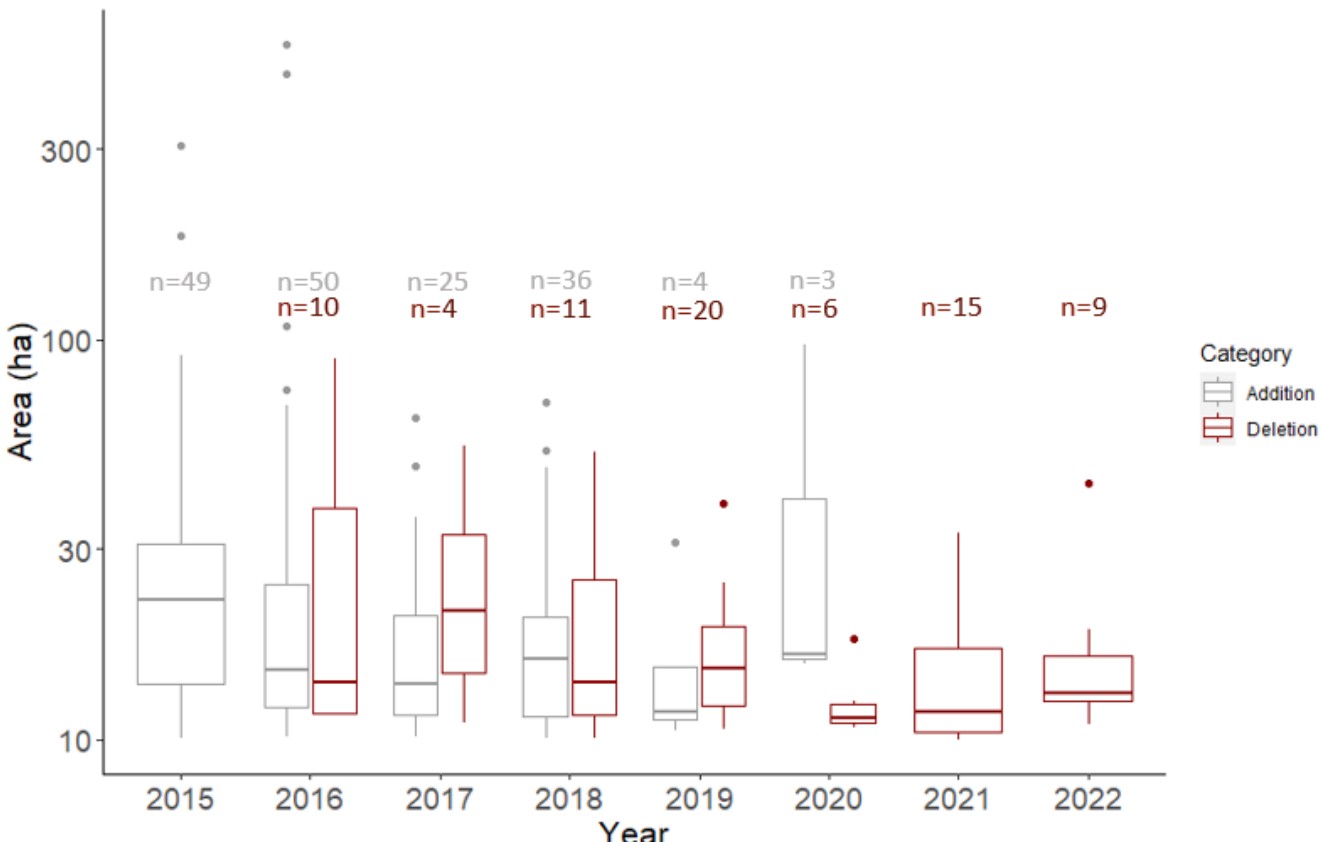

**Figure 5.** Boxplots of the area of wildfires which were added and removed (in the refining process) compared to the original INPA database. The term n is the number of wildfires that were added/removed to each year in the unified database. The years 2021 and 2022 are not shown in the addition category, as we did not identify wildfire events which were missing in the INPA database.

3.1.2. Improving the Temporal and Spatial Accuracy of the Database

In terms of spatial accuracy, and based on low mean RDNBR values and a visual check, 75 wildfire events were added by splitting 1 large wildfire event in the original INPA database into several adjacent wildfires that occurred on different dates. In terms of the temporal accuracy of the wildfire's occurrence date, we found that the date ranges of 77 (3%) wildfires between 2015 and 2022 were completely incorrect (errors ranged from 1 day to 91 days, Figure 6) according to the calculation of RDNBR index and its rank, as well as visual checking (Figure A1, stage 1). A non-significant, weak, and negative linear correlation was found between the difference between incorrect and correct wildfire occurrence dates and wildfire size (Figure 6). In addition, 1569 wildfire date ranges were shortened based on the RDNBR rank of the mean RDNBR value in a time series of the relevant season for each wildfire polygon (Figure A1, stage1). Table 2 details the number and percent of the rank of the RDNBR index calculated from the Landsat-8 and Sentinel-2 time series. Most wildfires (79–98%, depending on the year) had an index rank between 1 and 3 that matched their occurrence time, whereas 0–12% of wildfires had an index rank larger than 3 (which indicates low wildfire severity), and 1–17% of wildfires had negative RDNBR index (which indicates regrowth or unburned areas).

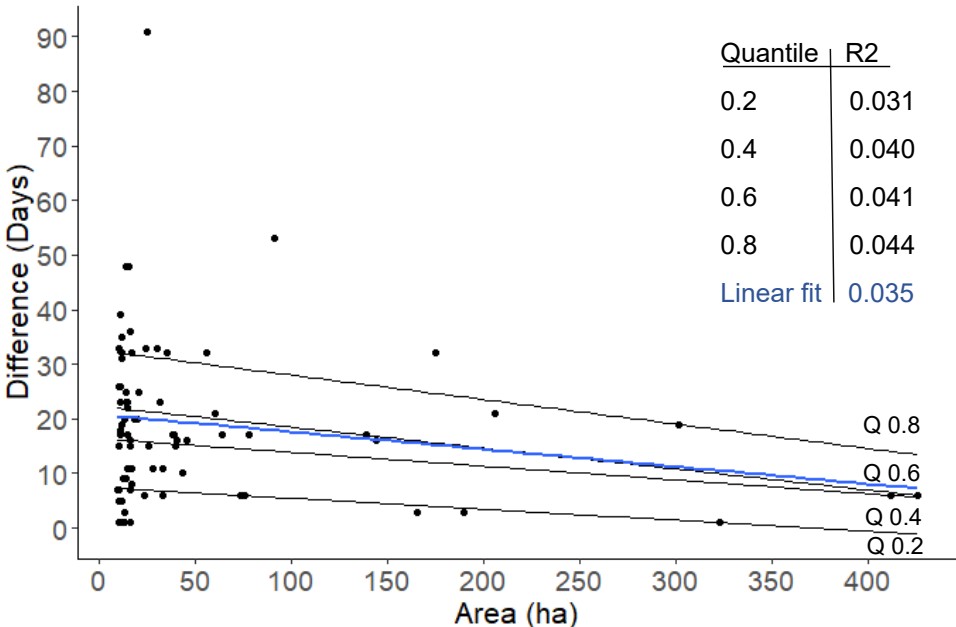

**Figure 6.** Absolute differences between incorrect wildfire dates in the original database (INPA) and the revised dates in the unified database, as a function of wildfire size (n = 77).

**Table 2.** RDNBR rank index from Landsat 8 (L8) and Sentinel 2 (S2) time series for the wildfire season in Israel (end of April to November) from 2015 to 2022. The percentage was calculated according to the final (merged) number of INPA total wildfires each year. The number in parentheses indicates the total number of wildfires that had a high RDNBR value expressed in rank value between one to three, a rank value larger than three, and a negative index value (which indicates regrowth or unburned areas).

| Year | Number of Wildfires | L8 RDNBR Rank 1–3 | L8 RDNBR Value < 0 | L8 RDNBR Rank > 3 | S2 RDNBR Rank 1–3 | S2 RDNBR Value < 0 | S2 RDNBR Rank > 3 |
|---|---|---|---|---|---|---|---|
| 2015 | 211 | 93% (196) | 2% (4) | 5% (11) | | | |
| 2016 | 303 | 93% (282) | 6% (17) | 1% (4) | | | |
| 2017 | 158 | 97% (154) | 1% (2) | 1% (2) | | | |
| 2018 | 218 | 82% (178) | 17% (36) | 0% (1) | 79% (173) | 11% (25) | 9% (20) |
| 2019 | 374 | 84% (313) | 15% (57) | 1% (4) | 93% (348) | 4% (14) | 3% (12) |
| 2020 | 370 | 86% (317) | 6% (22) | 5% (20) | 95% (353) | 2% (7) | 3% (10) |
| 2021 | 451 | 91% (411) | 8% (38) | 0% (2) | 94% (424) | 2% (10) | 4% (17) |
| 2022 | 191 | 98% (188) | 2% (3) | 0 | 87% (167) | 1% (2) | 12% (22) |
| Overall | 2276 | 90% | 7% | 2% | 90% | 4% | 6% |

### 3.1.3. Merging between Various Databases and Accuracy Assessment

The databases of MODIS active fire and the JNF had the lowest percentages (ranged from 7% to 21%) of joins made (i.e., many wildfires were missed by each of them), while VIIRS active fire and IFRS databases included more wildfire detections and thus more wildfires events were joined into the INPA updated database (ranged from 24% and 51%, respectively). Eventually, 1660 out of 2276 wildfires (73%) received an exact occurrence date (of them, 52% received an exact date from 1 source, 30%—2, 12%—3, 4%—4, 1%—5 different databases that had exact occurrence dates). Moreover, 987 (43%) wildfires received duration time from the IFRS database in addition to other attribute table data (Figure 1) from the various databases joined. The mean area of wildfires that received a duration time was 84 ha compared to the rest of the wildfires that had a mean burned area of 46 ha. Table 3

describes the degree of compatibility between the different wildfire databases compared with the refined database of the INPA (Figure A1, stage1). The spearman correlation coefficient between the number of databases joined (range from zero to four) per wildfire event from 2015 to 2022 and the area burned was 0.29 (Figure 7). The mean burned area of group 3 (3 different databases of 1 wildfire event were joined into the same wildfire event) was the highest, and was significantly different from the records with only 0–2; 4 wildfires were joined (ANOVA, F(4) = 87.9, *p*-value < 0.0001) (Figure 7). The burned area of wildfire events that occurred between 2015 and 2022 is presented in Figure 8, colored by the number of databases joined for each wildfire event. Overall, 1422 of the 2276 (62%) wildfires were identified in at least 2 different sources and had a high (1–3) RDNBR rank, while only 22 (1%) of the wildfires had low RDNBR rank (>3) and appeared only in a single database. When we compared the 2019 year with the global FireCCI database, we found that our database detected 201 more wildfires with additional burned area of 14,454 ha compared to the global database (Table A1). In addition, 19 wildfires appeared only in the global database; however, only 3 of them (which were located in croplands) had burned scars that were detected in Sentinel-2, while the rest were mistakenly classified as wildfires by FireCCI.

**Table 3.** Number and percent of wildfire events that were joined from each database source to the updated INPA database. The percentage was calculated compared to the INPA total wildfires number each year. The number in parentheses indicates the total number of matched wildfires events.

| Year | Number of Wildfires (INPA Original Database) | Number of Wildfires (INPA Updated Database) | IFRS | JNF | VIIRS | MODIS |
|---|---|---|---|---|---|---|
| 2015 | 162 | 211 | 47% (100) | 7% (14) | 43% (91) | 21% (45) |
| 2016 | 253 | 303 | 39% (117) | 7% (22) | 37% (111) | 16% (49) |
| 2017 | 133 | 158 | 51% (80) | 9% (14) | 29% (46) | 18% (28) |
| 2018 | 182 | 218 | 40% (87) | 17% (37) | 24% (52) | 11% (24) |
| 2019 | 370 | 374 | 42% (156) | 13% (48) | 38% (145) | 18% (67) |
| 2020 | 367 | 370 | 47% (175) | 12% (43) | 46% (170) | 15% (54) |
| 2021 | 451 | 451 | 43% (195) | 11% (49) | 48% (216) | 15% (67) |
| 2022 | 191 | 191 | 40% (77) | 10% (19) | 38% (72) | 14% (27) |
| Overall | 2109 | 2276 | 43% | 11% | 38% | 16% |

*3.2. Characteristics of the Unified National Wildfire Database*

3.2.1. Correlations between Wildfire Characteristics across Databases

The Spearman rank correlation coefficients between all quantitative wildfires' variables for all wildfire events >10 ha from 2015 to 2022 are presented in Figure 9. There were relatively high correlations among related variables from a single database, and relatively higher correlations between variables derived from VIIRS and MODIS active fires products compared to the other parameters. Similarly, there were relatively high correlations among DNDVI and NDVI indices from Landsat-8 and Sentinel-2 (r = 0.75 and r = 0.82, respectively), while the correlation between RDNBR indices was low (r = 0.32). Additionally, there was a relatively high and positive correlation between the burned area and the count of VIIRS active fire points (r = 0.45), while the correlation between wildfire duration and the burned area was lower (r = 0.36). When we tested the Spearman correlation between wildfire duration and burned area size for two groups, forest/grove and herbaceous vegetation, the correlation of herbaceous vegetation was higher (n = 528, r = 0.44, *p*-value < 0.001) compared to forest/grove wildfires (n = 55, r = 0.1, *p*-value = 0.5416). This analysis included a subset of the wildfires, covering only wildfires that had the same vegetation type classification from two independent sources (INPA and IFRS).

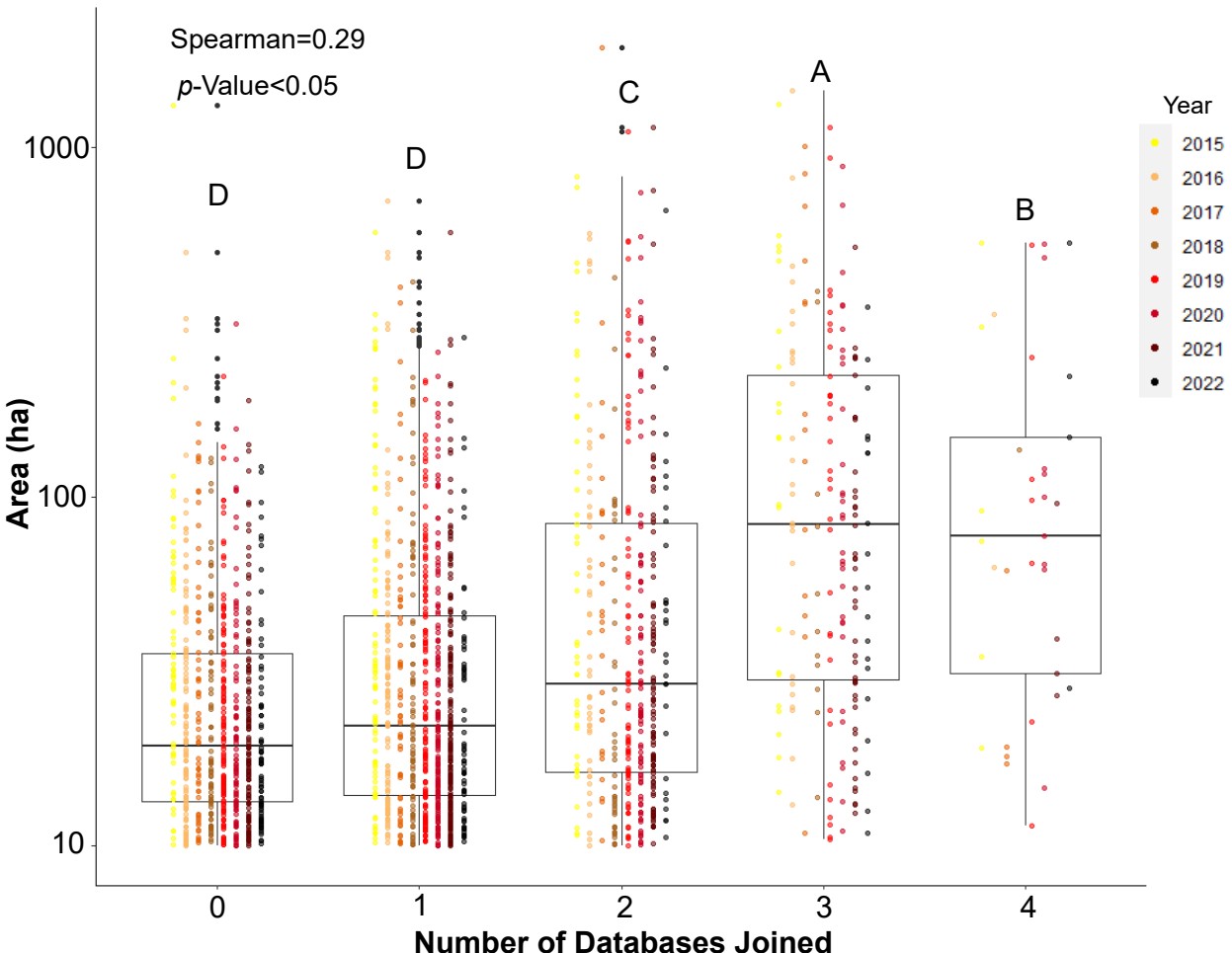

**Figure 7.** The number of databases joined to each wildfire event (2015–2022) and the area burned. Spearman's correlation coefficient denotes the correlation between the number of databases joined and area burned. Levels not connected by the same letter are significantly different (one way ANOVA).

3.2.2. The Temporal and Spatial Distribution of Wildfires (>10 ha) in Israel, 2015–2022

The mean and median burned area size and duration of wildfires in Israel are (63 ha, 4:40 h) and (24 ha, 3:20 h), respectively. Only 10 wildfires between 2015 and 2022 lasted more than 1 day. The largest wildfire (1450 ha) occurred at the end of November 2016 (and lasted four days) in the Jerusalem mountains and burned in an area characterized by groves and shrubland vegetation. Between 2015 and 2022, no clear trend was observed in terms of area burned or wildfire numbers across years (Figure 10). The correlation between the sum of the area burned and the number of wildfires for the eight-years period (n = 8) and by month (n = 63) was relatively high, at (r = 0.69, *p*-value = 0.053) and (r = 0.87, *p*-value < 0.001), respectively. When we analyzed the relationship between mean burned area and month by year (Figure 11), we found a known seasonal pattern of wildfire events in Israel, in which the largest wildfires tended to occur in transition months (April–May and October–November). The frequency of wildfires from 2015 to 2022 and the INPA land cover mapping for 2019–2020 is presented in Figure 12. A higher fire frequency was typical in herbaceous vegetation in military training zones.



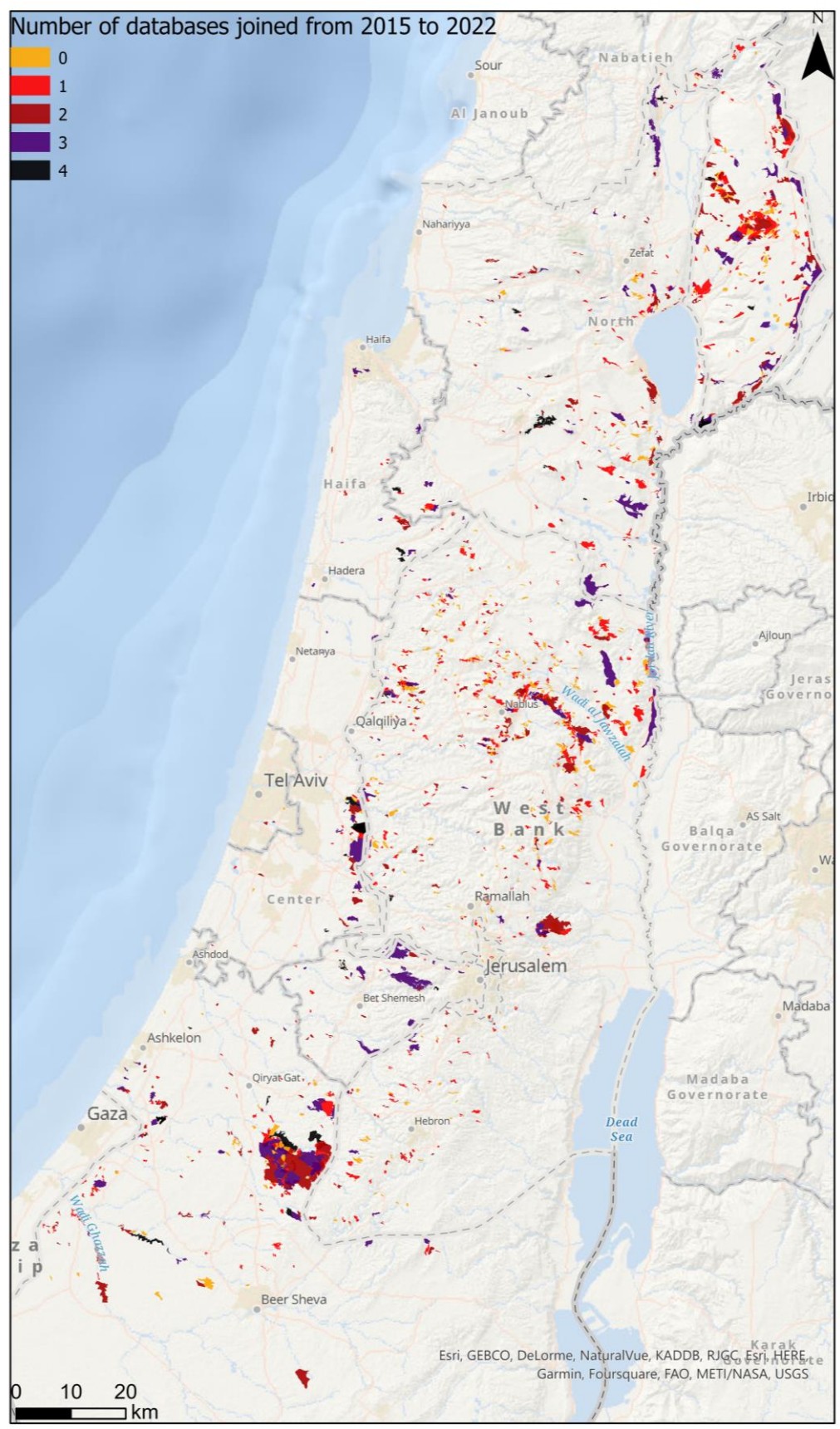

**Figure 8.** Wildfire scars from 2015 to 2022 colored by number of databases joined to INPA database.

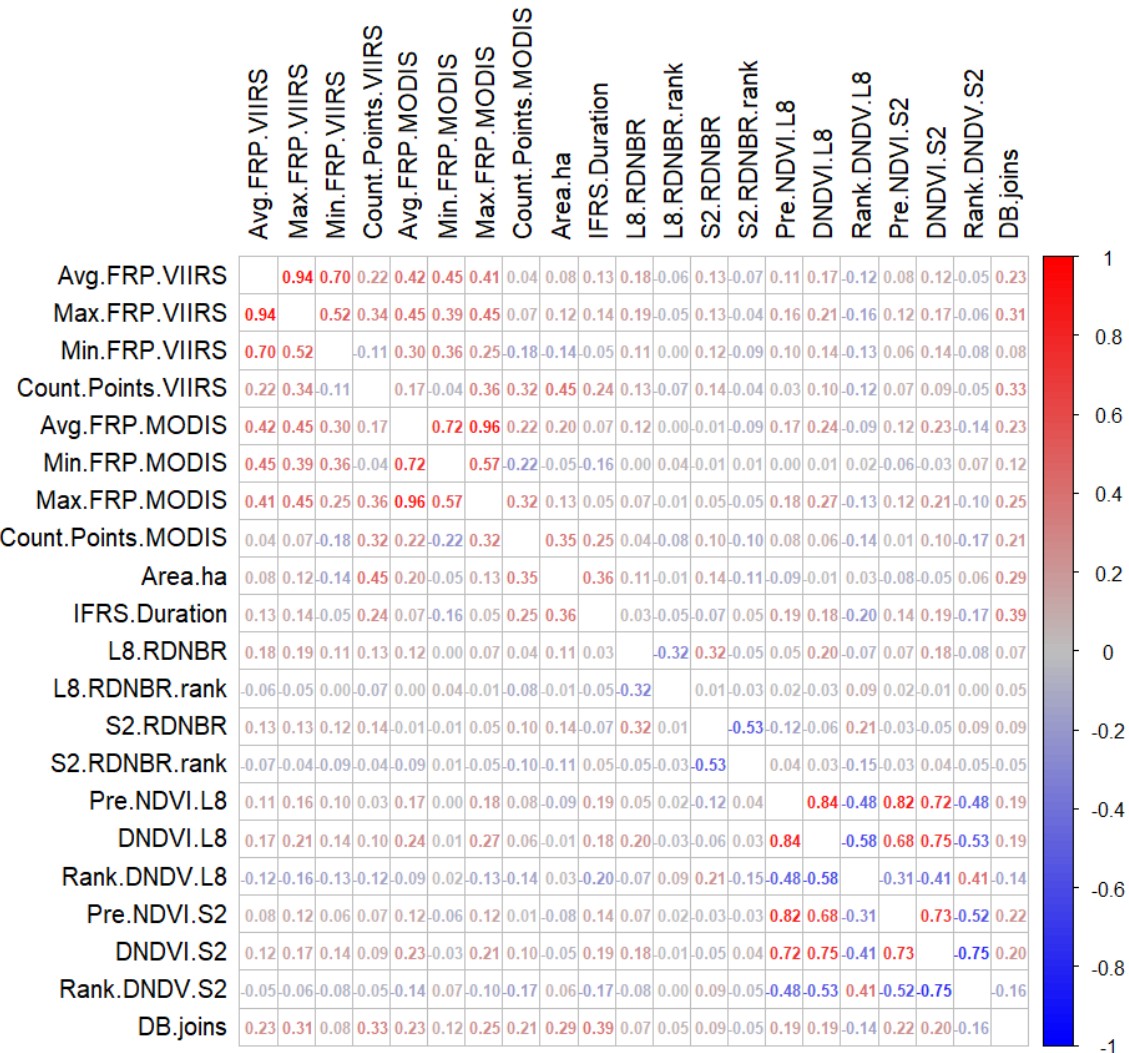

**Figure 9.** Spearman's rank correlation coefficients between the quantitative variables of the wildfires in the unified database (n = 2276) (area > 10 ha).

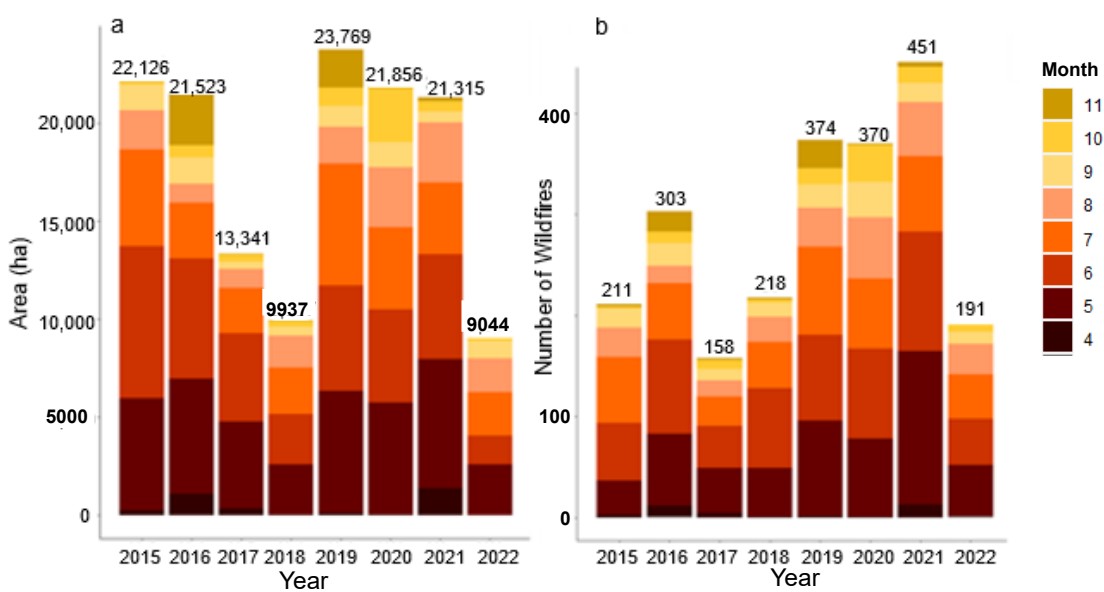

**Figure 10.** Total burned area (**a**) and number of wildfires (**b**) by month and year.

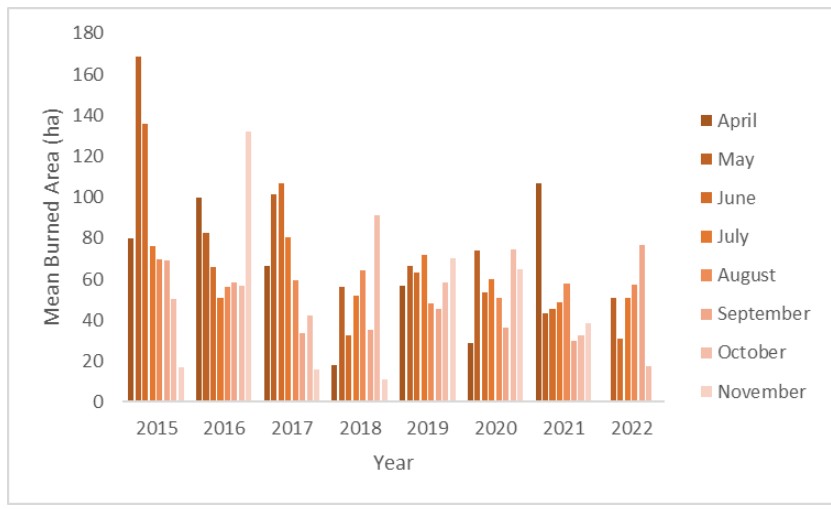

**Figure 11.** Mean burned area by month and year.

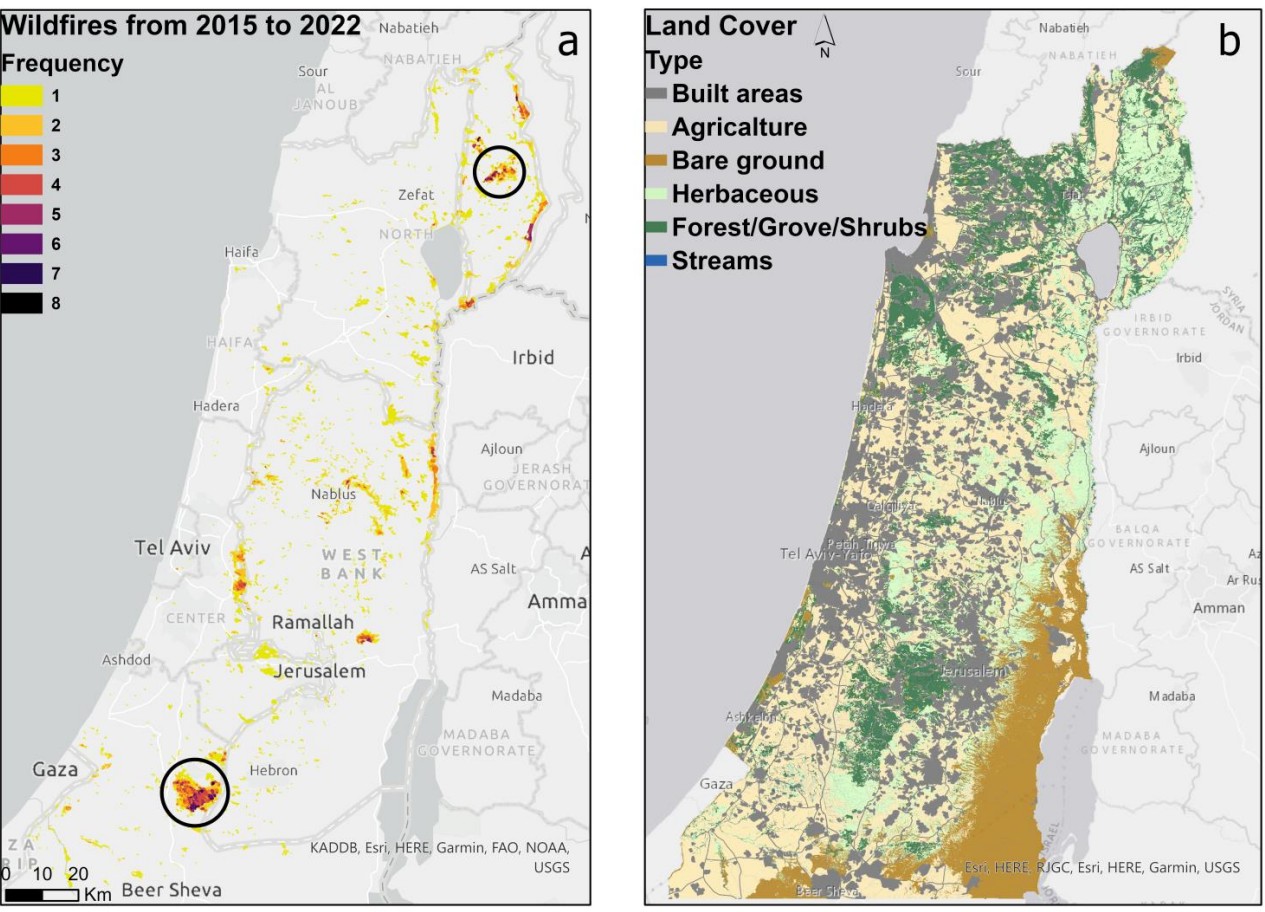

**Figure 12.** The frequency of wildfires from 2015 to 2022 (**a**) and land cover 2019–2020, INPA (**b**). Circles indicate two examples of military training zones.

## 4. Discussion

Effective fire risk assessment requires a thorough understanding of the wildfire regime in a region, which in turn relies on having a robust database of wildfire history at both local and regional scales [63–65]. However, in many regions and countries, the assembly of information into fire atlases is fragmentary and inconsistent, which results in recurring problems of completeness and spatial and temporal inaccuracies. Here, we present a

methodology for generating a national fire database which we exemplify in Israel, but suggest that the lessons we learned while designing this methodology can also be applied elsewhere, especially if the fire regime is characterized by a predominance of relatively small fires, and where data collection efforts are inconsistent across stakeholders.

### 4.1. Incomplete Records of Wildfire Events, Spatial and Temporal Inaccuracies, and Limitations

To construct a complete database of wildfires, there is a need to define the minimal burned area (minimum mapping unit) that will be included in the database. This fundamental requirement (which is not restricted to the case of mapping wildfires [66,67]) stems from the limitations of contemporary remote sensing approaches in terms of detection of small fires. In turn, the wildfire size distribution in the database should reflect as closely as possible the actual wildfire distribution: most of the wildfires are small, few are large, and large wildfires account for the most damage and area burned [61,68]. We found that 10 ha was the minimal burned area that we could reasonably detect using multiple independent approaches (both field-based and remotely sensed), and which at the same time corresponded to the actual wildfire size distribution in Israel, which is similar to the size (6.3 ha) found in Lebanon (Israel's neighboring country to the north, which has a similar climate [45]).We stress that the choice of minimal burned area threshold is not straightforward and reflects a tradeoff between detectability and outcome. On the one hand, mapping smaller wildfires is a challenge, as it requires a better spatial resolution coupled with short revisit times (to pinpoint ignition dates). On the other hand, small wildfires do not influence ecosystems as strongly as large wildfires, which cause major social, economic, and ecological costs [69,70]. This focus on outcome enables the choice of a burned area threshold that is not extremely small and thus technically unfeasible.

In developing our methodology, we found that calculating the mean RDNBR index (a common index for mapping fire severity) and its rank provided a useful automatic tool for fixing temporal inaccuracies in wildfire occurrence date ranges, with a high percentage (~90%, Table 2) of agreement between a high RDNBR rank in a time series (1–3 highest values) and the wildfire occurrence date range. We assume that this step, however, will be more beneficial in forested areas which are characterized by a relatively consistent spectral signal across the wildfire season and have a high disturbance after wildfires [71] compared to wildfires in agricultural and herbaceous areas, as these are characterized by high variability in surface reflectance throughout the growing season and faster recovery times [72]. In addition, this method detected only major spatial inaccuracies of fire scars (Figure A2), whereas smaller spatial inaccuracies in fire scars (which are the result of the manual digitization from satellite images) were not detected or fixed. In the case of a small country such as Israel, it is possible to manually map fire scars by visual digitization of satellite images. However, even wildfires larger than 10 ha could still be accidentally missing when manually mapped by people, but the probability of detection increases with wildfire size. Hence, it is essential to develop automatic and reliable mapping methods [16,73–75]. Such methods should distinguish between wildfires in different land cover/ecological units and different RDNBR thresholds because the results of using the same burn severity threshold for mapping fire scars (in the Israeli example) are not good enough and require additional manual refinement.

Creating an accurate, detailed, and reliable wildfire database relies mainly on the availability of data from both remote sensing and field data, coupled with the quality of these datasets. We found that when there was only one field data source for a given wildfire, the reliability of the information in terms of occurrence date or wildfire duration was unknown, as it was collected manually and is therefore prone to more biases and mapping errors. However, while in the original INPA database we could not assess the reliability of the data, in the new developed national database we were able to rate the accuracy of a given wildfire event with its characteristics and potential commission errors by the number of independent databases that included the same wildfire event, combined with the value of remote sensing spectral indices. In order to assess the accuracy of a spatial

dataset derived from remote sensing, one needs another dataset which is supposed to have better spatial and temporal accuracy [76]. However, for building this national database of wildfires we used all available datasets of wildfires in Israel, and globally available datasets of wildfires have a coarser spatial resolution [32,33]. Following our analysis, we expect that a few small-medium (>10 ha) wildfires each year could still be missing, in addition to spatial and temporal inaccuracies of wildfires that appeared only in one database, probably due to the problematic nature of manual data collection.

### 4.2. The Importance of a Detailed Database

Fires have complex dynamics, as the relationships among vegetation, climate, human activities, and topographical components are nonlinear and convoluted, and change across different regions and initial conditions [7]. Our analysis (Figure 9) reflected this complexity, as most of the numeric variables we extracted from various databases were not strongly correlated. For example, the spearman correlation found between wildfire size and wildfire duration was relatively low (r = 0.36); however, when we tested this correlation only for herbaceous vegetation, the correlation increased to r = 0.44 compared to only forest wildfires (r = 0.1). Additionally, a detailed and complete database of the wildfires history has the potential value for further research on improving wildfire public safety and the wildland–urban interface [77]. Hence, in order to provide a thorough characterization of a given fire event, it is essential to quantify various variables from multiple independent data sources [78]. This can be achieved only by combining remote sensing sources with field data collected by stakeholders, as remotely sensed data and field data are often complementary and synergistic in their description of fire characteristics.

### 4.3. Spatial and Temporal Distribution of Wildfires in Israel

Although we reconstructed only a relatively short period in Israel's fire history (8 years), which cannot be used to assess wildfire trends, our database (Figures 10 and 11) aligns with the known pattern of the wildfire size distributions by month and year in Israel. Most fires occur in the summer, but the largest fires (in terms of area) tend to occur in transition seasons [46]. During this period, the synoptic systems that cause major forest fires are the North African ('Sharav') cyclone and the Red Sea trough system, as they both carry hot and dry air [50]. There is an especially high probability for large fires in the fall, when fuel moisture is especially low after a few months without precipitation [45,79]. The ability of the unified database to reflect these patterns confirms it general validity, opening the door to future analyses that may deliver novel insights about the fire regime in general, and, as importantly, providing information for wildfire management and policy both at local and national scales. The database we developed will be made available to all relevant agencies in Israel to improve coordinated actions for wildfire management in Israel.

### 5. Conclusions

Creating a unified database is a crucial step for achieving a better understanding of a region's fire regime. The combination of remote sensing with field-based data collected by land management agencies is critical since these organizations collect additional complementary information which cannot be obtained from remote sensing alone. In this research, we presented methods to overcome the various problems of wildfire databases. First, to overcome the incompleteness in databases, it is important to decide on the minimum wildfire size that reflects the actual wildfire size distribution, i.e., most of the wildfires are small and few are large, while large wildfires contribute to the most area burned. Second, it is crucial to fix temporal and spatial errors in existing databases, especially when data are collected manually and as such are prone to various types of human errors. We found that calculating the mean RDNBR index and its rank in a time series around a given fire event was an automatic and reliable tool for fixing temporal inaccuracies, as well as spatial inaccuracies in cases where one large wildfire consisted of several different wildfires that occurred in different dates. In general, as wildfires are a complex phenomenon there is a

need to describe them using various independent variables from different sources. Hence, it is important to combine both field data and remote sensing data to construct a complete, accurate, reliable, and detailed database at both local and regional scales. In future work, we will use this database for retrospective analysis of the effectiveness of the Meteorological Fire Risk Index that is used by the Israel Meteorological Service and test physically based models of live fuel moisture content (LFMC) estimations for Israel.

**Author Contributions:** Conceptualization, E.G., N.L. and A.B.-M.; methodology, E.G., N.L. and A.B.-M.; software, E.G.; validation, E.G.; formal analysis, E.G.; investigation, E.G.; resources, E.G.; data curation, E.G.; writing—original draft preparation, E.G.; writing—review and editing, A.B.-M. and N.L.; visualization, E.G.; supervision, N.L. and A.B.-M. All authors have read and agreed to the published version of the manuscript.

**Funding:** This research received no external funding.

**Institutional Review Board Statement:** Not applicable.

**Informed Consent Statement:** Not applicable.

**Data Availability Statement:** The database will be provided upon request.

**Acknowledgments:** This work was supported by "Atid Baivrit" and David Amiran Scholarships of the Hebrew University of Jerusalem, Israel. We thank Israel Nature and Parks Authority, Jewish National Fund, and Israel Fire and Rescue Services for their wildfires data providing.

**Conflicts of Interest:** The authors declare no conflict of interest.

## Appendix A

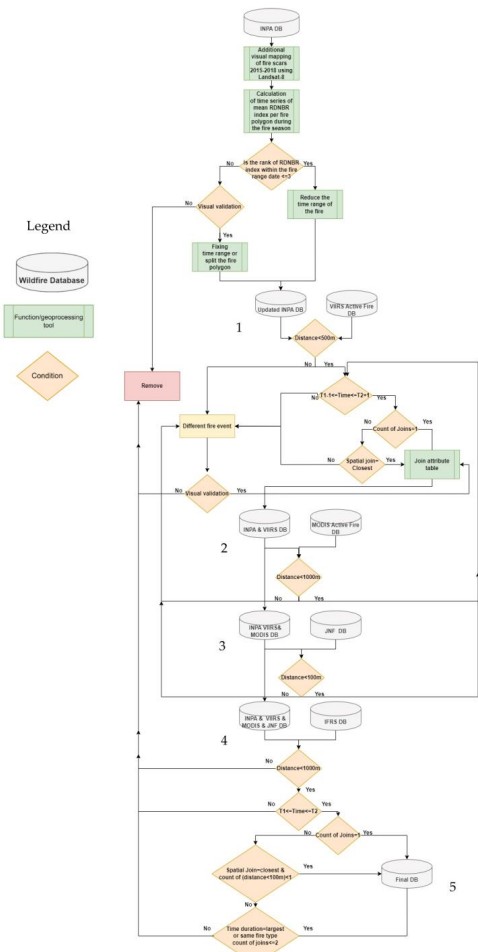

**Figure A1.** Flowchart of the creation process of Israel's national wildfires database.

    (1)    Refining the INPA database from Landsat-8 and Sentinel-2 manual digitization.
    (2)    Join the VIIRS active fire database with refined INPA database.
    (3)    Join the MODIS active fire database with INPA and VIIRS databases.
    (4)    Join the JNF Wildfire database with INPA and VIIRS and MODIS databases.
    (5)    Join the IFRS wildfire database with INPA and VIIRS and MODIS and JNF databases.

### Appendix B

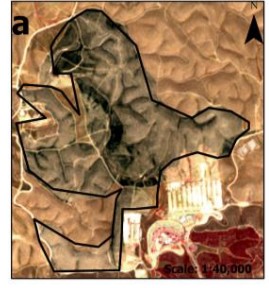 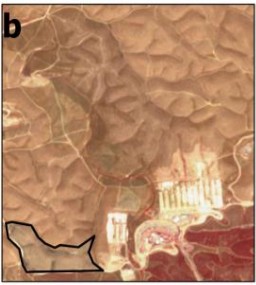 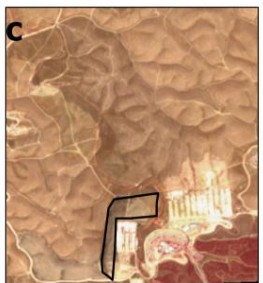 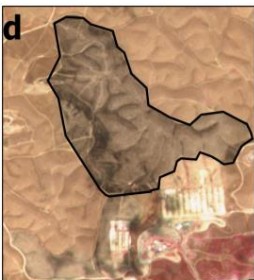 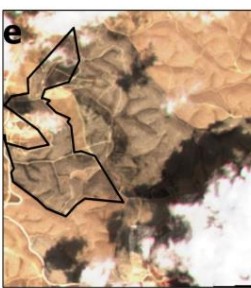

**Figure A2.** Example of a fixed spatial inaccuracy. (**a**)—the original mapping from the INPA database; (**b**–**e**) fixed mapping. (**b**)—11 May 2018; (**c**)—21 May 2018; (**d**)—26 May 2018; (**e**)—31 May 2018. (The dates indicate the date of the first sentinel-2 imagery (shown in a false color composite) in which the wildfire scar was detected.)

### Appendix C

**Table A1.** The number of wildfires and the sum area burned by month, from the global database (FireCCIS310), and our developed national database.

| Month | Wildfires Number FireCCIS310 | Wildfires Number National DB | Burned Area (ha) FireCCIS310 | Burned Area (ha) National DB |
|---|---|---|---|---|
| 4 | 5 | 2 | 108 | 114 |
| 5 | 35 | 94 | 1936 | 6228 |
| 6 | 49 | 85 | 3066 | 5374 |
| 7 | 38 | 87 | 2185 | 6238 |
| 8 | 8 | 39 | 690 | 1868 |
| 9 | 15 | 23 | 457 | 1050 |
| 10 | 5 | 16 | 158 | 938 |
| 11 | 18 | 28 | 715 | 1957 |
| Total | 173 | 374 | 9314 | **23,769** |

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
