# Peer review of "Constructing a Comprehensive National Wildfire Database from Incomplete Sources: Israel as a Case Study"

_fire, doi:10.3390/fire6040131_

Round 1

Reviewer 1 Report

The basic science of this paper is conducted in a good way and is of appropriate standard. The author and other co-authors wrote this paper according to journal scope and modern trends. They tried to improve the original dataset by identifying the missing fire and correcting the spatial and temporal range of the fine. I am glad to review this paper but there is no novelty in this paper. There are four big issues and I hope the author can try to partly or fully solve them.

The first issue is that the author did not assess the accuracy of the updated result. The authors showed the big gap between the original dataset (manual work) and updated dataset. But I also care whether there are some errors rate in the updated dataset. If the error rate in the updated dataset is much lower than the original one. I will confirm the improvement. The author should consider whether there are some ways to check both errors in both two dataset and the old one has poorer detection accuracy than the new one.

The second concern is how to link the indices (remote sensing) with the real burn severity which will influence the threshold setting. Currently, the author tried to identify the burning with the ranking. There are still GIS work. I recommend the Key and Benson’s work (http://gsp.humboldt.edu/OLM/Courses/GSP_216/labs/rmrs_gtr164_13_land_assess.pdf) or Ponomarev’s work (https://www.mdpi.com/2571-6255/5/1/19)

With real burn point data (There are some reports from the ranger).and the burn idex data for that point, I think the threshold can be set more reasonably. Or you have other ideas about how to connect the real data, please explain it in your resubmission.

The third one is that how author tell the difference among the drought, fire and insect. Because the drought and fire might also cause a low NDVI and other indices. Please explain how you exclude these disturbances in your updated version.

The last concern is that the whole Israel has multiple type of sub-land cover like mountainous areas, coastal areas, and interior range land. Each of them has their own characteristics. I hope the authors could consider the fire characteristics and their thresholds separately. At least the author can explain them in the discussion and the method part.  

The following will focus on specific problems line by line.

1.      Introduction part, please also mention some research about the field work analysis on fire.

2.      Line 85-87 Authors can list the algorithm in the method part.

3.      2.1 Study areas please use some quantitative way to describe your study areas like the areas of your study site, the composition of each land cover, annual precipitation, and your temperature. The numbers can make more sense to the reader.

Reviewer 2 Report

Overall, this manuscript presents a well-structured and organized study on the development of a methodology for constructing a comprehensive national wildfire database in Israel. The study highlights the importance of combining field data and remote sensing data to overcome the limitations and incompleteness of existing wildfire databases.

The methodology presented is thorough and includes detailed steps for data collection, preprocessing, and analysis. The study also addresses some of the common issues in existing wildfire databases, such as spatial and temporal errors and incomplete coverage.

The results of the study show that the developed methodology is effective in constructing a comprehensive wildfire database for Israel, with information on the occurrence date, fire duration, ignition cause, and more for wildfires larger than 10 ha. The authors also provide a blueprint for developing large-scale and comprehensive fire databases elsewhere, which will be useful for researchers and land management agencies worldwide. The manuscript is well-written and the figures and tables are clear and concise.

However, there are a few minor issues that need to be addressed.

1.      For example, the manuscript could benefit from additional details on the statistical methods used for data analysis and validation. Also, while the study focuses on the case of Israel, it would be interesting to discuss the potential limitations and challenges of the methodology in other regions with different wildfire regimes and land use practices.

2.      The methodology describes the process of identifying missing wildfires by re-examining Landsat 8, Sentinel-2 images and manually mapping the extent of those wildfires. However, it is not clear how the authors determined which wildfires were missing from the original INPA database. The methodology does not provide details on the criteria used to identify missing wildfires and how the accuracy of the manual mapping process was assessed.

3.      Moreover, the methodology relies on visual inspection of pre-fire and post-fire images to identify wildfires in the INPA database. This approach has several limitations, including the subjective nature of visual inspection and the potential for missing smaller wildfires. The methodology does not discuss how these limitations were addressed or the potential impact on the accuracy of the resulting wildfire database.

4.      Finally, the methodology focuses on wildfires larger than 10 ha, which creates a bias towards larger fires and may result in incomplete coverage of the wildfire regime. The authors do not discuss the potential impact of this bias on the accuracy of the wildfire database or provide a justification for the chosen threshold

Therefore, I recommend that the authors revise the methodology section to provide more details on the criteria used to identify missing wildfires and the accuracy of the manual mapping process. The authors should also address the limitations of visual inspection and the potential impact of the chosen threshold on the completeness and accuracy of the resulting wildfire database.

Overall, this manuscript is a valuable contribution to the field of wildfire management and provides useful insights for developing comprehensive wildfire databases in other regions. Therefore, I recommend it for publication after addressing the minor issues mentioned above.

Reviewer 3 Report

The goal of this research was to develop a methodology for creating a comprehensive national wildfire database. As a case study, the Authors created and tested this methodology for Israel. The methodology combines data on Israeli wildfires from two sources: remote sensing and field data collected by government agencies, and it covers the years 2015 to 2022. The manuscript is well-structured, clear, relevant to the field, and scientifically sound. The results of the manuscript are reproducible based on the details provided in the methods section. However, all figures with charts could be larger because they couldn't be read. In addition, I believe the paper should include more information about their future work in the conclusion. 

Reviewer 4 Report

The authors have co-constructed a comprehensive national database on forest fires from a case study in Israel.

The article is relevant to the journal and has a rigorous methodology.

Some aspects that may help to improve the quality of the article are:

1. With what data reliability the article has been made knowing that the data base on forest fires is often incomplete in terms of spatial and temporal coverage, as well as in terms of documentation of fire outcomes.

2. The methodology developed is rigorous ; it combines data on forest fires in Israel from two sources: remote sensing and field data collected by government agencies. There are many variables in studies such as date of occurrence, fire duration, cause of ignition, etc.  With what level of confidence and with what error the authors can ensure that the presented methodology facilitates future studies on forest fire risk by providing a robust and unified database of Israel's fire history from 2015.

3. How have the authors decided to set the minimum size of forest fires to reflect the actual distribution of forest fire size?

4. justify the weighting given to both field and remotely sensed data to build a complete, accurate and detailed database at both local and regional scales?

5. Enrich the state of the art with the following references.

-Li, D., Cova, T. J., Dennison, P. E., Wan, N., Nguyen, Q. C., & Siebeneck, L. K. (2019, October 1). Why do we need a national address point database to improve wildfire public safety in the U.S.? International Journal of Disaster Risk Reduction. Elsevier Ltd. https://doi.org/10.1016/j.ijdrr.2019.101237

-Short, K. C. (2014, January 3). A spatial database of wildfires in the United States, 1992-2011. Earth System Science Data. https://doi.org/10.5194/essd-6-1-2014

Round 2

Reviewer 1 Report

The authors have followed my comments and improved the paper approximately. I agree to accept it.